# Revealing the Influences of Sex Hormones and Sex Differences in Atrial Fibrillation and Vascular Cognitive Impairment

**DOI:** 10.3390/ijms22168776

**Published:** 2021-08-16

**Authors:** Ya-Ting Chang, Yung-Lung Chen, Hong-Yo Kang

**Affiliations:** 1Department of Neurology, Kaohsiung Chang Gung Memorial Hospital, Chang Gung University College of Medicine, Kaohsiung 83301, Taiwan; emily0606@cgmh.org.tw; 2Department of Psychiatry, Osaka University Graduate School of Medicine, Suita 565-0871, Osaka, Japan; 3Section of Cardiology, Department of Internal Medicine, Kaohsiung Chang Gung Memorial Hospital, Chang Gung University College of Medicine, Kaohsiung 83301, Taiwan; feymanchen@gmail.com; 4Graduate Institute of Clinical Medical Sciences, Chang Gung University College of Medicine, Kaohsiung 83301, Taiwan; 5Center for Hormone and Reproductive Medicine Research, Department of Obstetrics and Gynecology, Kaohsiung Chang Gung Memorial Hospital, Chang Gung University College of Medicine, Kaohsiung 83301, Taiwan

**Keywords:** sex differences, sex hormones, atrial fibrillation, vascular cognitive impairment, androgen receptor, estrogen receptor

## Abstract

The impacts of sex differences on the biology of various organ systems and the influences of sex hormones on modulating health and disease have become increasingly relevant in clinical and biomedical research. A growing body of evidence has recently suggested fundamental sex differences in cardiovascular and cognitive function, including anatomy, pathophysiology, incidence and age of disease onset, symptoms affecting disease diagnosis, disease severity, progression, and treatment responses and outcomes. Atrial fibrillation (AF) is currently recognized as the most prevalent sustained arrhythmia and might contribute to the pathogenesis and progression of vascular cognitive impairment (VCI), including a range of cognitive deficits, from mild cognitive impairment to dementia. In this review, we describe sex-based differences and sex hormone functions in the physiology of the brain and vasculature and the pathophysiology of disorders therein, with special emphasis on AF and VCI. Deciphering how sex hormones and their receptor signaling (estrogen and androgen receptors) potentially impact on sex differences could help to reveal disease links between AF and VCI and identify therapeutic targets that may lead to potentially novel therapeutic interventions early in the disease course of AF and VCI.

## 1. Introduction

Sex hormones are steroid hormones that bind to sex hormone receptors; they are also referred to as sex steroids, gonadocorticoids, and gonadal steroids. Androgens and estrogens—the main sex hormones—act differently in males and females. Hormonal effects are primarily mediated by rapid nongenomic actions through membrane-associated receptor signaling cascades and by slow genomic actions via classical sex steroid receptors [1,2].

17β-Estradiol, also referred to as E2, is the most potent and prevalent form of estrogen; it is synthesized mainly in granulosa cells of the female ovaries and male Sertoli cells. Estrogen synthesis also occurs locally in the central nervous system from cholesterols or is converted from aromatizing androgens in presynaptic terminals [2]. Testosterone is converted to E2 via P450 aromatase in the hypothalamus of men, where mental/social sex determination occurs [3]. E2 exerts its physiological effects by activating various estrogen receptors (ERs), which have at least three forms: ERα, ERβ, and membrane-bound G protein-coupled ER (GPR30/GPER1) [4]. ERα and ERβ are well-studied nuclear steroid receptors that are associated with the cytoplasm, plasma membrane, and nucleus in vascular smooth muscle cells, cardiomyocytes, and vascular endothelial cells in the mammalian cardiovascular system [5,6,7]. Both types of ER function as ligand-activated transcription factors and therefore exert long-term genomic effects by modulating gene expression through direct interaction with highly conserved DNA-binding domains of nuclear ERs and estrogen response elements located near the promoter or enhancer regions of estrogen-targeted genes [4]. GPR30/GPER1, which is highly expressed in the hypothalamic–pituitary–adrenal axis, has been shown to act as a membrane ER mediating the nongenomic effects of E2 [8]. GPR30/GPER1 signaling has been shown to improve spatial memory, possibly via neurotransmitter release and generation of new spines on hippocampal neurons [8]. Moreover, GPER1 activation leads to the phosphorylation of the classical intracellular ERα, suggesting that crosstalk with ERα contributes to anxiety and social behaviors, such as social memory and lordosis behavior, in mice [8].

Testosterone, the principle androgen, is mainly synthesized in male testicular Leydig cells and female ovarian theca cells and secreted into the blood stream. It is converted into a more potent androgen, dihydrotestosterone (DHT), by 5α-reductase in the testes and prostate (in men), ovaries (in women), skin, and other parts of the body. Both androgens serve as ligands for androgen receptor (AR), a ligand-dependent transcription factor and a member of the nuclear receptor gene superfamily that mediates androgen signaling in males and females [9]. Upon binding to testosterone or DHT, AR undergoes conformational changes to recruit several essential co-regulators, translocates into the nucleus, and regulates the actions of genomic androgen by interacting with androgen response elements (AREs) located near the promoter or enhancer regions of androgen-targeted genes [1]. Numerous AR co-regulators play vital roles in AR stability and transcription, which influence proteasome degradation and affect their ligand and DNA-binding capabilities [1]. AR is expressed in several vascular cell types, such as smooth muscle cells, endothelial cells, and blood cells including macrophages and platelets [10,11]. Several physiological regulators of cardiovascular function, such as nitric oxide release, Ca^2+^ mobilization, vascular apoptosis, hypertrophy, calcification, senescence, and reactive oxygen species generation, are influenced by nongenomic androgen actions [11].

In the heart and brain of both males and females, sex hormones regulate the structure and function of cardiovascular and neural systems to modulate behavior and disease patterns at distinct molecular and cellular levels via the actions of sex hormone receptors [12]. Sex differences have been observed in diseases such as atrial fibrillation (AF) and vascular cognitive impairment (VCI). These sex differences and differential responses to sex hormones in the diseases of heart and brain, which influence cardiovascular and cognitive functions, were previously considered as separate events. Now, emerging evidences indicate that hormonal communications between the heart and brain occurs partly through the cerebral vasculature, where sex hormone signaling may act differently in male and female via the genomic and nongenomic actions of sex hormone receptor.

## 2. Sex Differences in AF

### 2.1. Sex Differences in the Epidemiology and Clinical Outcomes of AF

Evidence shows that the incidence of AF is higher in men than in women. Women with AF are older and have a higher prevalence of hypertension, valvular heart disease, and heart failure (HF) with preserved ejection fraction and lower prevalence of coronary artery disease than men with AF [13,14]. Women with AF were found to be more often symptomatic than men, with greater symptom severity [13,14]. In previous studies, women with AF had significantly higher rates of life-threatening adverse events (e.g., acquired long QT syndrome with class Ia or III antiarrhythmic drugs) [15,16] or sick sinus syndrome requiring pacemaker implantation [17] following rhythm control with antiarrhythmic drugs than men. Women with AF were less likely to undergo electrical cardioversion and experience more delayed AF catheter ablation referral than men, possibly reflecting the occurrence of AF later in life among women [14,18,19,20]. Although catheter ablation showed better results than drug therapy for reducing the incidences of AF and stroke [21,22,23], women may have less favorable outcomes [19,20], with higher rates of procedure-related complications [19], than men. Women were more likely to undergo atrioventricular nodal ablation for AF than men [24]. Previous studies also showed that women had increased hospitalization rates due to AF recurrence after AF ablation but were less likely to undergo repeat ablation or cardioversion [25,26].

### 2.2. Pathophysiology of Sex Differences in AF

The pathophysiology of AF development is associated with autonomic neural control and electrical and structural remodeling [27]. The mechanisms of sex differences in AF may involve structural, electrophysiological, and cardiac autonomic modulation and neuro-humoral responses [28].

#### 2.2.1. Structural Remodeling

Although healthy women have relatively smaller left atria than men, women referred for AF ablation had larger left atria, possibly due to older age, longer AF duration, and more comorbidities [29,30,31]. Cardiac fibrosis is critical for AF development, with the more pronounced fibrosis in women possibly being associated with TGFβ/Smad3 pathway upregulation and older age [32].

Epicardial fat has also been correlated with higher AF prevalence, progression to atrial fibrosis, permanent AF, and even higher recurrence rates after catheter ablation [33,34]. The menopause-associated reduction in estrogen levels causes increased epicardial fat, upregulation of related signaling pathways, and fibrotic remodeling [35]. Epicardial adipocytes can release proinflammatory adipokines and activate chemotactic monocyte chemoattractant protein-1/C-C chemokine receptor 2 pathways that promote inflammatory macrophage accumulation. The crosstalk between adipocytes and inflammatory cells depends on the release of cytokines interleukin (IL)-1, IL-6, and tumor necrosis factor-α (TNF-α) by fat tissue macrophages [36]. Other proinflammatory adipokines such as leptin and resistin are also associated with incident AF in women [37].

#### 2.2.2. Electrical Remodeling

Mouse studies showed that male cardiomyocytes had greater late sodium current, calcium transients, and sarcoplasmic reticulum calcium contents in the left atrial posterior wall than female cardiomyocytes, possibly contributing to increased ectopic activity [38]. The pulmonary veins of male mice had faster spontaneous beating rates, greater burst firing, and more delayed afterdepolarizations than those of female mice [39]. Cardiomyocyte calcium and sodium channels are differentially regulated by sex hormones, which explains the differences in the action potential period (APD) between male and female mice [40,41,42,43,44,45,46,47,48,49]. The shortened APD in male atria may be proarrhythmic by facilitating re-entry, whereas the longer APD in female atria may exert antiarrhythmic effects against AF in contrast to its proarrhythmic effects in the ventricles.

Biochemical and histological analyses of atrial tissue acquired during cardiac surgery revealed that men and women with AF exhibited generally similar remodeling-induced changes in connexins and collagen; however, women exhibited stronger AF-induced increases in Cx40 expression [32]. mRNA expression analysis of genes encoding ion channel subunits that are important in cardiac conduction and arrhythmogenesis of left atria from explanted human hearts revealed differences in remodeling according to sex, with lower expression levels of transcripts encoding K(v)4.3, KChIP2, K(v)1.5, and K(ir)3.1 in failing female left atria than in male left atria [50].

#### 2.2.3. Autonomic Neural Control and Neuro-Humoral Modulation

The autonomic nervous system, composed of the sympathetic and parasympathetic systems and the intrinsic neurohormone network, is critical for AF pathogenesis [51,52,53] and is involved in the initiation and maintenance of AF. The parasympathetic system contributes to AF principally by shortening APD and increasing the dispersion of refractoriness in the atrial myocardium, facilitating the initiation and maintenance of AF [54]. Vagal activation exerts these effects mostly via acetylcholine-activated Kt channels [55]. Sympathetic stimulation can also promote AF by increasing Ca^2+^ release and influencing the conductive properties and refractoriness of cardiac tissue, causing afterdepolarization formation, inducing AF [56]. Assessment of heart rate variability showed that compared with similarly aged men, women appeared to be vagal activity-dominant [57]. However, low estrogen and elevated progesterone levels lead to increased catecholamine levels, and sympathetic activity is higher in the luteal phase of the menstrual cycle [58]. These sex differences in autonomic neural control disappear with aging considering the decreased estrogen levels in menopausal women [59].

## 3. Sex Differences in VCI

### 3.1. Sex Differences in the Epidemiology and Clinical Outcomes of VCI

Cerebrovascular disease (CVD), the second-most common cause of cognitive impairment (CI) and dementia, frequently contributes to cognitive decline in neurodegenerative dementia. VCI is associated with vascular disorders that may coexist with neurodegeneration [60,61,62] and includes milder forms of CI and vascular dementia (VaD). Many patients with CVD develop several cognitive disabilities. Some studies suggested that male sex is a risk factor for CI [63,64]; others found that female sex is predictive of the increased risk of CI [65,66]. Although dementia disproportionally affects females, there are conflicting findings on the influence of sex on the incidence and prevalence of VCI [67]. Sex-related differences in risk factors, cognitive profiles, rates of deterioration, pathogenesis, and outcomes remain unknown. Evidence has revealed a sex-specific pattern in the incidence of CVD, with women having lower incidence rates of both ischemic stroke and intracerebral hemorrhage (ICH) than men [68]. Among 860 patients with CVD, significantly more women than men had poor cognitive performance (approximately 15% difference) [67]. Despite the similar incidence of VCI between women and men [67], women tend to experience more severe strokes [69], whereas men frequently experience their first stroke earlier [69]. Risk factors for CVD such as AF, HF, myocardial infarction, high blood pressure, hyperlipidemia, obesity, and diabetes mellitus (DM) are more common among men [53,70,71]; however, the incidence rates of dementia associated with these risk factors are conflicting [53,70]. Some studies reported no significant difference in the risk for VCI between men and women [72,73,74], whereas others suggested that men had significantly higher incidences of VCI [75,76,77]. Studies have found that women experience poorer functional and cognitive decline after stroke than men [78,79,80]. Women had a greater risk for dementia among individuals with DM [70]. In a meta-analysis, sex differences in the prevalence of VCI were associated with age: VCI was more prevalent among men aged <79 but was more prevalent among women aged >85 [81].

Sex differences in the efficacy of stroke treatment have also been reported. Aspirin was found to be more effective in preventing stroke in women than in men [82], whereas warfarin was more effective for AF in men than in women [83]. Considering that therapeutic efficacy against stroke is implicated in the prognosis of VCI, the influence of sex differences is crucial in the clinical outcome of VCI. Sex differences also influence the efficacy of nonpharmacological interventions against VCI [84]. Thus, sex differences in the efficacy of stroke treatment should be determined.

As women tend to experience more severe stroke than men, they would have a higher incidence of VCI than men [85]. Within the first 3 weeks, the most important predictor of long-term functional outcome in patients with stroke is memory, which is associated with the medial temporal lobe (MTL) volume [86]. As men reportedly have larger MTLs than women [87], sex differences might affect the prognosis of VCI considering their influence on brain morphology. However, executive function was found to be a predictor of functional outcome and is associated with prefrontal volume [88]. The results regarding the influence of cognitive sex differences on VCI prognosis are inconsistent. Thus, the modulating effect of sex differences on the relationship between cortical volume and VCI prognosis remains unclear. Patients with VCI exhibiting memory, visuospatial, and executive impairments show significantly poorer global cognitive function, as assessed using the Mini-Mental State Examination (MMSE) [89]. Executive dysfunction, which can be measured using the Trail-Making Test A, was demonstrated to be a predictor of the modified Barthel index in patients with VCI [89].

Acetylcholinesterase inhibitors (AChEIs) can improve cognitive function in patients with VCI [90]. Cholinergic augmentation led to significant improvements in MMSE scores after 4 weeks in patients with post-stroke CI and VCI [90]. The neural system and cholinergic pathways, which comprise the basal forebrain, substantia innominata, striatum, cerebral cortex, some brainstem nuclei, and spinal motor neurons [91], are vulnerable to vascular damage, which can cause CI. It has been suggested that AChEIs modulate CI by compensating for the lack of intracerebral cholinergic neurotransmitters by inhibiting acetylcholine hydrolysis. This has been considered an effective treatment pathway in patients with post-stroke CI and VaD [92]. Sex differences in pharmacological effects have been associated with higher sensitivity to the toxic effects of organophosphate cholinesterase inhibitors in males [93]. Therefore, older males and females might respond differently to AChEIs because of either sex-specific differences in the structure and function of the cholinergic system, pharmacokinetics, memory function, or the effects of aging or AD on such processes [93].

### 3.2. Pathophysiology of Sex Differences in VCI

#### 3.2.1. Sex Differences in Brain Structure and Function among Individuals with VCI

To determine the influence of sex differences on VCI, the pathogenesis of stroke [69], cerebral infarction [69], intracranial hemorrhage [69], efficacy of secondary prevention [82,83], and risk factors for cerebral atherosclerosis should be considered [53,70,71], along with structural and functional sex differences in the brain [94]. Regional sex differences in brain volume might be implicated in sex-specific CI during VCI.

Sex differences have been demonstrated in several cognitive tasks. Men have been reported to outperform women in spatial ability [95], whereas women outperform men in verbal ability [96]. Cognitive sex differences have been associated with differences in structural and functional brain organization.

While men have higher metabolism within the temporal-limbic areas, women have higher metabolism in the cingulate areas [97]. Men experience increased functional connectivity (FC) within and among parietal-occipital regions, as evaluated using resting-state functional magnetic resonance imaging, whereas women experience increased FC within and among frontotemporal regions [98,99]. Moreover, men have stronger inter-network FC, whereas women have stronger intra-network FC [100].

Sex differences also contribute to variability in brain morphology. Men have significantly larger frontal, temporal, left parietal, and insula areas than women [101]. Women exhibit a higher gray matter (GM)/white matter (WM) ratio in the parietal cortex [102,103], cingulate gyrus [87,104], and insula [104]. Men have increased GM volumes in the MTL and entorhinal cortex, whereas women have increased GM volumes in the right inferior frontal and cingulate gyri [87]. After correcting for whole-GM volume, women exhibited greater GM percentages in the dorsolateral prefrontal cortex and superior temporal gyrus than men [105], implying that women have better language-related abilities than men [105]. Regarding WM structures, women have significantly lower fractional anisotropy in the right deep temporal regions [106].

#### 3.2.2. Sex Differences in Risk Factors for VCI

Although risk factors for CVD such as DM, obesity, and hypertension are more common in men [53,70,71], they more adversely affect women [71]. However, hyperlipidemia, MI, AF, and HF show higher influence in men [71]. While men are more likely to experience stroke than premenopausal women, similar incidences of stroke have been recorded between men and postmenopausal women [69,107]. Sex differences according to the type of stroke have also been reported: brain infarctions and ICHs are more common in men, whereas subarachnoid hemorrhages are more common in women [69].

Women are more prone to obesity and obesity-related DM, which increases the risk of VCI [108,109]. Therefore, sex differences in the effects of type 2 DM on VCI suggest that women are more adversely affected than men. More women are overweight or obese after the age of 45 years, whereas more males are overweight at a younger age. Besides age, the influence of sex differences on body mass index (BMI), body fat distribution, brown adipose tissue, metabolic syndrome, and adipokines leads to an increased risk of DM and DM-associated VCI in women [109].

Obesity is another important risk factor for VCI. The effect of BMI on VCI is more pronounced in women than in men. Higher midlife BMI has been associated with increased vascular risk factors, changes in adipokines (plasminogen activator inhibitor-1, IL-6, TNF-α, angiotensinogen, adiponectin, and leptin), and brain structure alteration [110], whereas lower BMI later in life has been associated with neurodegenerative processes [111].

Besides DM and obesity, sex differences also affect hyperlipidemia. Decreased high-density lipoprotein (HDL) and increased triglyceride levels in men have been associated with an increased risk for all-cause dementia [112]. In women, low HDL levels have been associated with increased WM lesions and silent brain infarcts [113]. Large vessel strokes (macroangiopathy and arteriosclerosis), small vessel disease (microangiopathy and arteriolosclerosis), and microhemorrhages are the main causes of VCI [114]. Therefore, the lower HDL levels in women may explain their higher risk for VCI. Given that the genetic effects of APOE4 are associated with lipoprotein metabolism, studies have found that higher levels of APOE4 allele are associated with a higher risk of VCI [115,116].

## 4. The Interactive Relationship of Sex Hormones and Sex Differences between AF and VCI

Current data show that patients with AF are prone to stroke, dementia, HF, and increased mortality [117,118]. Epidemiologic studies reported that ischemic stroke was a significant risk factor for dementia and that the characteristics of stroke, incident medical illnesses associated with cerebral hypoxia or ischemia, and older age also affect this risk [119,120,121]. While cerebral hypoperfusion, vascular inflammation, cerebral small vessel disease, and several risk factors have been associated with AF and VCI [122], nevertheless, emerging evidence indicates potential mechanisms may uncover how sex differences, hormone levels, and hormone receptor signaling influence the development and progression of AF-related VCI, including changes in anatomy, pathophysiology, disease onset and incidence, disease severity, disease outcome, and response to treatment (Figure 1).

### 4.1. Effect of Sex Differences on Outcomes in Patients with AF-Related VCI

Our previous study showed that among patients with AF aged >55 years, women had a greater risk of dementia than men [123]. However, the impact of female sex on the risk of developing dementia in patients with AF can vary according to different dementia types. Women have greater risk for Alzheimer’s disease than men among patients with AF aged >55 years, whereas no sex differences were noted in their risk for developing VaD [123]. Furthermore, a Korean research group reported that catheter ablation better maintained and even improved the cognitive function of patients with AF compared with drug therapy [124]. However, the effect of catheter ablation on the clinical outcomes, especially dementia, between men and women remains unclear.

### 4.2. Pathophysiology of Sex Differences in AF and VCI

Previous studies showed that sex might be associated with atrial amyloidosis [125,126], which is more common in women, particularly older women [126]. This may be associated with increased atrial natriuretic peptide expression following ER stimulation in the presence of E2. Accordingly, elevated atrial natriuretic peptide levels promote amyloid formation and deposition and cause atrial fibrosis and AF, which may cause thrombosis and stroke [125,126]. Amyloidosis has been considered an important etiology of dementia [127]. A previous study showed that female hormone supplementation increased the risk of stroke in patients with AF [128]. Macrophage-produced cytokines, including IL-l, IL-6, IL-12, and TNF-α, were also reported to be associated with atrial fibrosis and AF attacks [129]. Sex differences have been shown to influence macrophage-related inflammatory processes [130]. The aforementioned studies therefore suggest a strong correlation between sex differences, AF, stroke, and dementia. However, no study has yet confirmed the exact causes and mechanisms.

## 5. Effect of Sex Hormone Deficiency/Excess on AF and VCI

### 5.1. Sex Hormone Deficiency/Excess and AF

#### 5.1.1. AF and Androgen Signaling

Serum testosterone levels decline with age [131], and an estimated 39% of men aged >45 years have hypogonadism [132]. Epidemiological data on the association between AF and testosterone are conflicting. The Framingham study revealed an association between the incidence of AF and reduced total testosterone levels in men aged ≥55 years [133]. Another smaller cross-sectional study demonstrated a similar association between reduced testosterone levels and AF [134]. The FINRISK study indicated that low testosterone levels were associated with an increased risk of future AF and/or ischemic stroke in men but were protective in women [135]. In contrast, the Multi-Ethnic Study of Atherosclerosis showed that higher levels of endogenous bioavailable testosterone appeared to contribute to AF development [136]. The differences in these findings may be partially attributed to methodological differences in testosterone measurement (total vs. bioavailable testosterone) and competing mechanisms of direct and indirect testosterone effects. Previous studies showed that the acute effects of testosterone are beneficial and differ from the chronic effects of testosterone exposure [137]. Cardiac L-type calcium channels are crucial for maintaining intracellular calcium homeostasis and are therefore essential in inducing arrhythmia [138]. Chronic exposure of rat cardiomyocytes to testosterone (24–30 h) increased L-type calcium channels and the frequency of calcium sparks without increasing sarcoplasmic reticulum calcium load. Conversely, acute treatment of cardiomyocytes with testosterone led to a decrease in L-type calcium channels. These differences were attributed to genomic androgen pathway activation mediated by nuclear AR in chronic treatments and the direct blocking effects via nongenomic androgen signaling in acute testosterone treatments [137,138]. Androgen levels have also been related to the incidence of AF in patients with congenital or acquired diseases, which might cause chronic androgen excess or deficiency. Klinefelter syndrome is the most common male sex chromosomal disorder and is characterized by small testes, azoospermia, and increased luteinizing hormone and follicle-stimulating hormone levels [139]. Data from Korean National Health Insurance Service indicated that patients with Klinefelter syndrome without a history of AF and ischemic stroke had higher incidences of AF, but not stroke, than the control group [140]. The lack of androgen and decreased diastolic function in patients with Klinefelter syndrome could explain the causal relationship between Klinefelter syndrome and AF. Androgen deprivation therapy with abiraterone for metastatic prostate cancer was associated with increased incidences of atrial tachycardia, HF, hypokalemia, hypertension, and edema associated with abiraterone-induced hypermineralocorticism [141]. According to data retrieved from the Danish Registry Cohort Study, women with polycystic ovary syndrome (PCOS) had a twofold increased risk for AF than those without PCOS [142], potentially due to insulin resistance and elevated BMI among those with PCOS.

#### 5.1.2. AF and Estrogen Signaling

Although premenopausal women have lower incidences of AF than men, such incidences are more frequent after menopause, particularly among women aged >50 years. This phenomenon suggests that the protective effects of estrogen and/or the harmful effects of prominent loss in estrogen during menopause on AF [143]. This representation may be associated with the effects of AF-associated risk factors, such as hypertension, dyslipidemia, and metabolic syndrome, which were also elevated after menopause and increased the incidence of AF [144]. The prevalence of AF in pregnant women is 0.05%, and it usually occurs in those with structural heart disease among whom AF incidences were marginally higher (approximately 1.3%) [145,146]. Patients with preeclampsia also presented with increased progesterone wave duration and dispersion and atrial electromechanical coupling interval, as measured using tissue Doppler echocardiography [147]. These are well-known markers for increased AF incidence. The incidence of AF during the peripartum period has been mainly associated with drug therapy, such as terbutaline during tocolysis, or peripartum cardiomyopathy [148,149,150]. ERα and ERβ, the two main types of nuclear ERs, are highly expressed in the heart [10], are abundant in cardiomyocyte mitochondria, and regulate mitochondrial function [151]. The conduction properties of cardiomyocytes are also directly affected by estrogen. Chronic estradiol treatment showed a modulatory effect on coronary artery smooth muscle potassium channels and cardiac calcium channels [152]. Estrogen is also critical for excitation and contraction coupling considering that it regulates calcium homeostasis in the heart and membrane density and L-type Ca^2+^ channel expression in cardiomyocytes [153,154,155,156]. E2 inhibits the occurrence of early afterdepolarization and the ectopic trigger activity induced by depolarization, potentially serving as an antiarrhythmic drug [156].

### 5.2. Sex Hormone Deficiency/Excess and VCI

#### 5.2.1. VCI and Androgen Signaling

Clinical studies have shown that appropriate testosterone and DHT balance is important for improving the outcomes of stroke in men. Low testosterone and DHT levels have been associated with increased risk and severity of stroke, mortality, increased infarct size, and poor stroke outcomes in men [157,158]. Conversely, testosterone replacement therapy in men aged >65 years was shown to increase the incidence of CVD events [159]. This could be attributed to the association between androgens and vasoconstriction, reduced vasodilatation, and increased vasoconstrictors [160,161,162]. Another mechanism for the effects of testosterone administration could involve leptin levels. Increased testosterone levels have been associated with lower leptin levels [163]. As increased leptin levels have been associated with reduced infarct volume and neurological deficits in rodent models of ischemic stroke, increased testosterone level in the elderly might be implicated in poor stroke outcomes [164]. Considering that increased severity of neurological deficits with stroke is an important predictor of VCI, androgen levels have been implicated in the risk and prognosis of VCI [165]. Several studies showed that testosterone was beneficial for brain function because it prevented neuronal cell death, balanced brain oxidative stress and antioxidant activity, improved synaptic plasticity, and increased cognitive function [166]. Promising associations have been found between the decline in cognitive function and low testosterone levels [167].

Besides the effects of androgen levels, the androgen signaling pathway has been associated with VCI pathogenesis after stroke. To facilitate neuroprotection, the androgen signaling pathway suppresses the Toll-like receptor 4/nuclear factor kappa B signaling pathway, subsequently alleviating microglia inflammatory responses [168]. Moreover, the androgen signaling pathway can regulate amyloid precursor protein metabolism and reduce β-amyloid production [169] and has been implicated in the CREB and MAPK/ERK signaling pathways: the former pathway improves the hippocampal synaptic structure, and the latter enhances neuroviability [170].

AR activation has been suggested to protect intact male mice from memory impairments caused by aromatase inhibition [171]. While the aromatase inhibitor letrozole blocked memory in only gonadectomized males, suggesting that circulating androgens or hippocampal androgens increased due to aromatase inhibition may support memory consolidation in intact males, males whose AR was blocked by the antagonist flutamide showed impairment of memory consolidation [171]. Various lengths of CAG (glutamine) repeat polymorphism in AR have been associated with cognition in older men [172]. Several studies have demonstrated that although E2 is necessary for inducing long-term potentiation, DHT is necessary for inducing long-term depression of synaptic transmission in the hippocampus [171,172]. This contribution was proven by administering sex hormones in rodent models and using agents that block their synthesis or specific receptors. The general opposite role of sex hormones in synaptic plasticity is apparently dependent on their local availability in response to low or high frequencies of synaptic stimulation, thereby inducing bidirectional synaptic plasticity. Investigation of the effects of AR expression on brain function and cerebrovasculature will provide additional insight into the potential mechanisms of novel therapeutic approaches for VCI.

#### 5.2.2. VCI and Estrogen Signaling

Although men are more likely to experience stroke than premenopausal women, this disparity subsides after menopause. This may be explained by loss of E2 during menopause. The relationship between ovarian sex hormone levels and stroke outcomes are supported by findings of altered outcomes over the estrous cycle, with smaller infarct size noted during proestrus (high E2) [173]. Other studies found that in women, both aging and ovariectomy exacerbate ischemia-/stroke-related outcomes [174,175]. The protective effects of E2 in experimental stroke models were reported in young male and female animals [176,177]. Studies have suggested that E2 has dose-dependent effects, such that E2 administration at physiological levels attenuates damage from experimentally induced stroke [178,179], whereas supraphysiological doses may be detrimental, given that they increase infarct size and aggravate oxidative stress, inflammation, and excitotoxicity [180,181]. While physiological levels attenuate injury from stroke [174,175], supraphysiological levels of E2 can be harmful; studies have reported increased infarct size, inflammation, excitotoxicity, and oxidative stress [180,181,182]. Age also interacts with the effects of E2: protective effects in young [179] but detrimental effects in old animals have been observed [179,183]. Because the increased severity of neurological deficits of stroke is an important predictor of VCI, E2 levels have been implicated in the risk and prognosis of VCI [165].

The effect of estrogen signaling on the brain is critical for the protective effects of E2 against excitotoxicity, inflammation, oxidative stress, and apoptosis [184,185,186,187,188]. The E2 signaling pathway can inhibit oxidative stress-induced and PARP1-dependent cell death via binding its alpha-type receptor [184]. It also promotes neurogenesis in rats by increasing hypoxia-inducible factor 1α and vascular endothelial growth factor expression [188]. Moreover, it suppresses inflammation by inducing a specific type of NMDA receptor by enhancing GRIA2 and NR2B expression [185]. As it maintains Bcl-2 expression, the E2 signaling pathway has been associated with the attenuation of ischemic injury-related CI [187]. Furthermore, E2 signaling influences Nrf2-ARE pathways in the hippocampus CA1 regions, thereby modulating CI relevant to cerebral ischemic changes [186].

Several ER subtypes have been identified, including ERα and ERβ [189]. In a study including 2625 women aged ≥65 years, Yaffe et al. obtained results that supported the association between ERα polymorphisms and the risk of CI [190]. *ERβ* gene polymorphisms (ESR2 rs4986938) were also associated with an increased risk for VaD in elderly Jewish women [191]. A specific ER subtype is also involved in VCI treatment and prevention [6]. An animal study showed that E2 enhances capillary density in the brain and primes tissue survival after experimental focal ischemia through ERα [192]. Further insight into the potential mechanisms of novel therapeutic approaches for VCI can be obtained by studying the effects of the expression of various ER subtypes on VCI/VaD-related brain regions and cerebrovasculature of young and aging females (e.g., those who are postmenopausal, ovariectomized, received hormone replacement, and reproductively senescent).

## 6. Effects of Sex Hormone Therapy on AF and VCI

### 6.1. Effects of Sex Hormone Therapy on AF

Studies suggest that antiestrogen treatment increases the incidence of AF, whereas estrogen-based hormone replacement therapy (HRT) decreases the risk of AF (Figure 2, left panel). However, the data from different studies are controversial [193,194,195]. While estradiol was shown to reduce the risk of AF, conjugated estrogens alone had been reported to increase the risk of AF [194,195]. Combined estrogen–progesterone-based HRT had no effect or decreased the incidence of AF [193,195]. These findings suggest that hormonal preparations and their ER specificity have complex interactions and effects. In addition, acute administration of E2 in postmenopausal women will prolong the conduction time in the right atrium and atrioventricular nodes, as well as the effective refractory period of the right atrium [196]. This result has been reproduced in a female mouse ovariectomized model, which leads to a shortened PR interval and the conduction time from the right atrium to the atrioventricular node, while estrogen replacement has the opposite effect [197]. A large cohort study involving 76,639 patients with low testosterone levels showed that those whose testosterone levels were normalized with testosterone replacement therapy had lower incidences of AF than those with low levels of testosterone and those who did not receive replacement therapy [198]. Moreover, participants who failed to attain normal total testosterone levels after testosterone replacement therapy had higher incidences of AF than those whose total testosterone levels were normalized following testosterone replacement therapy.

Data on the effects of testosterone replacement therapy are not consistent on different animal models of AF. An orchiectomized male Sprague Dawley rat model study confirmed the relationship between testosterone deficiency and AF [199]. The study suggested that the resolved electrically stimulated repetitive atrial responses after testosterone therapy were associated with decreased calcium leakage from the sarcoendoplasmic reticulum resulting from the normalization of the binding between FK506-binding protein and ryanodine receptor type 2. However, another study on aged rabbits showed the opposite effects, with testosterone replacement enhancing arrhythmogenesis in pulmonary veins and the left atrium, probably by enhancing adrenergic activity [200].

### 6.2. Effects of Sex Hormone Therapy on VCI

HRT generally has beneficial effects when initiated immediately after menopause and has detrimental effects when administrated later in life [201,202]. However, the effects of hormone receptor signaling on VCI are complex (Figure 2, right panel). Based on clinical studies on breast and prostate cancers, hormone therapy that blocks sex hormone production or suppresses hormone receptors to reduce or inhibit tumor growth in the breast and prostate was implicated with CI [203]. Considering that E2 and androgen are critical in regulating healthy brain and cognitive function [204], hormone therapy that blocks E2 and AR activity may be potentially harmful to patients with VCI. E2 and AR are both widely distributed throughout cerebral regions, particularly the hippocampus and prefrontal cortices, which are important for cognitive functions [205,206]. In postmenopausal women, HRT has been shown to attenuate CI in subjects with mild CI [207]. Data from animal studies demonstrate that HRT during the critical period is involved with neuroprotection [208]. In ovariectomized rats, HRT reduced the level of inflammatory and modulated neuroprotection process [209]. Animal studies showed that E2 has neuroprotective effects and regulates synaptic plasticity in the brain regardless of sex [4]. In males, testosterone is converted to E2 locally by aromatase. Therefore, testosterone may possibly exert neuroprotective effects and regulate synaptic plasticity in the male brain via E2, although whether testosterone has its own direct effect through AR remains unclear [210]. Because cognitive function does not rely on a specific brain region, it is determined by neuronal network interactions. Thus, understanding the neural mechanisms behind cognitive functions affected by sex hormones is valuable and warranted. It suggests that in postmenopausal subjects, HRT may be beneficial for decreased risk of VCI, but HRT administration during the critical timing is essential for attenuating progression of VCI.

## 7. Conclusions and Future Perspectives

With the current aging of the population, the prevalence of AF with VCI, including dementia, can be expected to reach epidemic proportions worldwide. Emerging evidence has showed that AF increases the risk of VCI via various mechanisms, albeit mainly through cerebral hypoperfusion and thromboembolism, which could cause silent cerebral ischemia. These direct consequences of AF on the brain might be consistent with other pathological factors such as tauopathies and plaque formation. These pathological factors are common among the elderly and reduce cognitive reserves and facilitate the development of dementia. Circulating sex hormone levels and proinflammatory biomarkers, particularly those related to endothelial dysfunction, might be involved in possible pathophysiological mechanisms for relationship between sex differences and atrial fibrillation, or between sex differences and VCI. The development of tools and instruments for the assessment of both conditions is an important issue. Further studies are warranted to understand the sex-specific effects of dementia risk factors on the incidence of AF and examine the underlying mechanisms of sex differences. The roles of sex hormone receptor pathways in patients with AF and VCI should be comprehensively investigated in longer, larger-scale prospective cohort studies with more accurate neuropsychological and cognitive function assessments. Additional clinical trials are needed to identify the best therapeutic approaches for preventing VCI progression in patients with AF. Moreover, deliberate stratification according to sex should be considered. An adequate sample size is needed to determine the therapeutic efficacy in men and women separately. The information presented herein may help establish new strategies for the development of individualized therapeutics and preventive medications for AF with VCI.

## Figures and Tables

**Figure 1 ijms-22-08776-f001:**
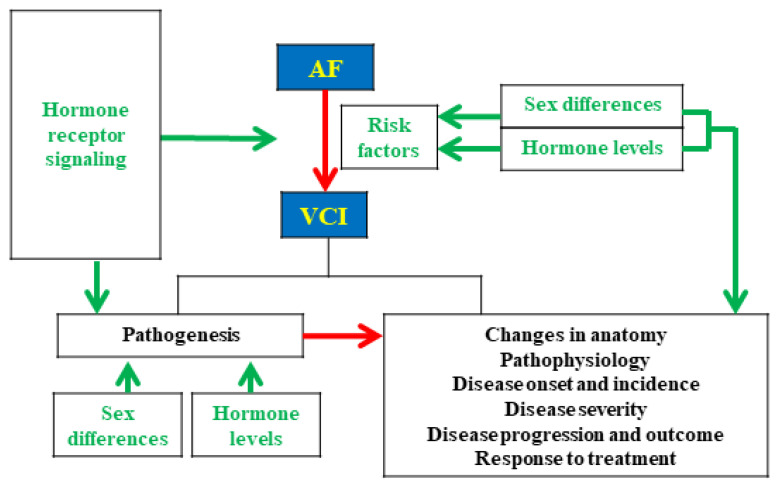
Hypothetic scheme indicates the potential links among sex differences, hormone levels, and hormone receptor signaling on the pathogenesis of atrial fibrillation (AF)-related vascular cognitive impairment (VCI).

**Figure 2 ijms-22-08776-f002:**
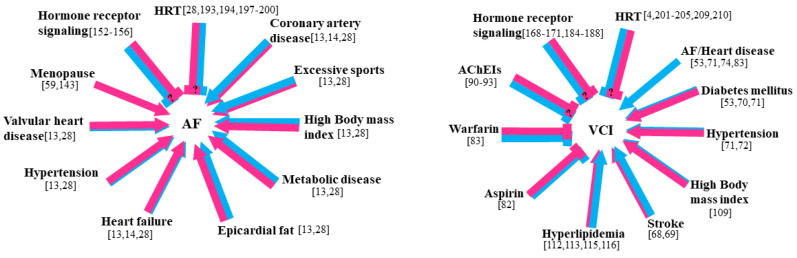
Schematic representation of sex differences in risk factors for atrial fibrillation (AF) and vascular cognitive impairment (VCI). Pink and blue colors indicate female and male risk factors, respectively. Risk factors that increase risk in both sexes are indicated by pink/blue arrows and the proportion of each color indicates which sex exhibits greater risk. The inhibitory symbol with a longer horizontal and a shorter vertical line is used instead of an arrowhead to indicate factors that may decrease risk. AChEIs = acetylcholinesterase inhibitors; HRT = hormone replacement therapy.

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
