# Peer review of "Revealing the Influences of Sex Hormones and Sex Differences in Atrial Fibrillation and Vascular Cognitive Impairment"

_ijms, 2021, doi:10.3390/ijms22168776_

Round 1
Reviewer 1 Report
In this review, the authors introduced the sex differences and the role of various sex hormones/ receptors signaling in AF and VCI, and discussed its potential involvement in the links between AF and VCI. Overall, it's a comprehensive review in an important area. I only have a few suggestions.
- As a review on sex differences and hormones, the authors should include more information on peri vs post menopause women and experimental animals with orchiectomy and ovariectomy.
- The part IV. The interactive relationship of sex hormones and sex differences between AF and VCI is hard to understand. The authors should revise this part to get it more logical and clear.
Author Response
Reviewer 1:
In this review, the authors introduced the sex differences and the role of various sex hormones/ receptors signaling in AF and VCI, and discussed its potential involvement in the links between AF and VCI. Overall, it's a comprehensive review in an important area. I only have a few suggestions.
1. As a review on sex differences and hormones, the authors should include more information on peri vs post menopause women and experimental animals with orchiectomy and ovariectomy.
Answers: Thank you for your suggestions. We have added more information on peri vs post menopause women and experimental animals with orchiectomy and ovariectomy as reviewer suggested. Please see page 11 (line 512-517) and page 12 (line 545-548 and line 556-558).
2. The part IV. The interactive relationship of sex hormones and sex differences between AF and VCI is hard to understand. The authors should revise this part to get it more logical and clear.
Answers: We apologize for any confusion. We have deleted several paragraphs to make it more logical as reviewer’s suggestion and hope that it is now clearer. Please see page 7 (line 296-298, 309-319) of the revised manuscript.

Reviewer 2 Report
The authors have done a good job of examining the relationship between sex differences or sex hormones and atrial fibrillation and vascular cognitive impairment (VCI). However, this paper seems to be lengthy and describes the same things. It is likely that sex hormones are involved in the mechanisms for relationship between sex differences and atrial fibrillation, or between sex differences and VCI. The authors should rewrite this paper with these in mind and focus on the main points.
Author Response
Response to the reviewers’ comments
The referees gave our paper positive comments by mentioning “…it's a comprehensive review in an important area.” However, reviewers also raised some concerns. Here are our responses to improve the manuscript, in light of the reviewer’s comments. (Point-by-point).
Reviewer 2:
The authors have done a good job of examining the relationship between sex differences or sex hormones and atrial fibrillation and vascular cognitive impairment (VCI). However, this paper seems to be lengthy and describes the same things. It is likely that sex hormones are involved in the mechanisms for relationship between sex differences and atrial fibrillation, or between sex differences and VCI. The authors should rewrite this paper with these in mind and focus on the main points.
Answers: Thank you for the suggestions. We agree with the reviewer that it is lengthy and redundant in section 4. Therefore, we have deleted several paragraphs to make it more logical as reviewer’s suggestion and hope that it is now clearer. We now revise the manuscript and put more focus on the main points that sex hormones are involved in the mechanisms for relationship between sex differences and atrial fibrillation, or between sex differences and VCI in the main text. Please see page 7 (line 296-298, 309-319) and page 12 (line 567-569) of the revised manuscript.
